# Cultivating Well-Being: An Exploratory Analysis of the Integral Benefits of Urban Gardens in the Promotion of Active Ageing

**DOI:** 10.3390/ijerph22071058

**Published:** 2025-07-01

**Authors:** Noelia Fernández-Salido, Alfonso Gallego-Valadés, Carlos Serra-Castells, Jorge Garcés-Ferrer

**Affiliations:** Research Institute on Social Welfare Policy (POLIBIENESTAR), University of Valencia, 46022 Valencia, Spain; alfonso.gallego@uv.es (A.G.-V.); carlos.serra@uv.es (C.S.-C.); jordi.garces@uv.es (J.G.-F.)

**Keywords:** urban gardens, older people, active ageing, holistic wellbeing

## Abstract

Ageing is a global demographic trend that has increased the total prevalence of multimorbidity, disability and frailty, posing ever greater challenges for public health systems. For older people, ageing is often associated with a loss of quality of life, independence and well-being. This study analyses the role of urban gardens as spaces that promote active ageing and contribute to the physical, psychological and social well-being of older adults. Focusing on the urban areaof Valencia, this research adopts a qualitative approach based on in-depth semi-structured interviews with 15 older adults who regularly participate in urban gardens. The findings indicate that urban gardens contribute significantly to active ageing by providing opportunities for regular physical activity, emotional well-being, social engagement, and improved nutrition through the cultivation of food by the participants themselves. These spaces also enhance autonomy, stimulate cognitive functions, elevate mood, and offer a renewed sense of purpose following retirement. Moreover, urban gardens serve as inclusive environments that promote intergenerational interaction and reinforce community bonds. As multifunctional spaces, they hold considerable potential for enhancing the quality of life among older adults and addressing key public health challenges associated with population ageing. Consequently, their integration into urban planning frameworks should be prioritised.

## 1. Introduction

Population ageing is one of the most significant demographic changes worldwide [1,2]. This phenomenon, which has become particularly relevant in recent decades, is closely linked to increased life expectancy and sustained decline in fertility rates [3,4]. As a result, demographic ageing has led to an increase in population multimorbidity, disability and frailty, posing growing challenges for public health systems and social services [5,6]. It is estimated that baby boomers will account for approximately 25% of the older population by 2030, which translates into an increase in mortality rates and the prevalence of chronic, musculoskeletal, lumbar and tumour diseases [7,8]. This study focuses on analysing how the regular participation of older people in urban gardens influences the promotion of active ageing and, therefore, their overall well-being, from a qualitative approach.

According to recent Eurostat [9,10], demographic projections for the EU-27 show accelerated growth in the oldest age group. Between 2024 and 2100, the percentage of people aged 80 or over is expected to increase 2.5 times (from 6.1% to 15.3%), and the percentage of people aged 65 or over is expected to increase from 21.6% to 32.5% [9]. Between 2019 and 2050, the number of people aged 85 and over is estimated to increase by 113.9%, from 12.5 million to 26.8 million, while the number of centenarians will reach 484,000 [10]. At the same time, it is projected that by 2050, more than 90% of European regions will have at least a quarter of their population aged 65 or over [11]. These projections reflect a profound demographic transformation that will have direct implications for healthcare systems, public policies and economic sustainability [12].

This demographic phenomenon is also reflected in health indicators. In 2023, life expectancy at age 65 increased in the EU by 0.7 years compared to 2022, with Spain recording the highest value (22.0 years) and Hungary the lowest (16.8 years) [4]. However, this longevity does not always translate into quality of life. In 2018, 68.5% of the adult population (aged 16 and over) in the EU-27 perceived their health as good or very good. This percentage falls to 47.8% among people aged 65 to 74, to 32.3% among those aged 75 to 84 and to 20.6% among those aged 85 and over [9]. At the same time, functional limitations also increase with age: in 2014, around 10.5% of adults aged 65 to 74 experienced limitations in walking, while 32.3% of those aged 75 and over reported severe difficulties [11]. This functional decline is linked, among other factors, to a higher prevalence of depressive symptoms in older people, which affected 13.1% of those aged 75 and over, compared with 6.5% in the 65–74 age group [9]. In Spain, life expectancy at birth in 2022 stood at 83.2 years—the highest in the EU—attributed to factors such as the Mediterranean diet, universal access to healthcare and efficient primary care [13]. Overall, this health, functional and emotional landscape highlights the need to develop strategies for active ageing, prevention of dependency and promotion of personal autonomy.

Old age is also a stage of life that is particularly vulnerable to social stereotypes, a phenomenon known as ‘ageism’. This concept encompasses a set of prejudices, assumptions and discrimination present in political, social and health systems, which systematically affect older people [14,15,16]. Recognised as the third form of discrimination in the world, after racism and sexism, age discrimination often constructs a negative and limiting image of older people as dependent, sick, unproductive, inflexible or fragile [17,18]. These representations not only influence social attitudes and institutional practices but also shape urban environments that often overlook the real needs and capabilities of the ageing population, reinforcing exclusion rather than supporting autonomy and participation. To address the consequences of global demographic change driven by population ageing—and the associated challenges to health, functionality, and overall well-being—it is essential to implement comprehensive strategies, policies, and programmes that foster a positive perception of ageing. This transformation involves leaving behind the traditional social stigmas surrounding old age, prevalent in many developed societies, and moving towards a paradigm focused on promoting active ageing [19,20]. The concept of active ageing was defined by the World Health Organisation in 1994 as the process of optimising opportunities for physical and mental health, social participation and security, with the aim of improving quality of life as people age. This definition recognises the human rights of older people and incorporates the United Nations principles of independence, participation, dignity, care and self-fulfilment [21,22]. More recently, initiatives such as the WHO Decade of Healthy Ageing 2021–2030 have reinforced the need to create environments that promote the well-being of older people through integrated actions in public health, urban planning and community participation [23]. The Age-Friendly Cities and Communities (AFCC) initiative, promoted by the WHO in 2007, has been implemented by multiple cities around the world to adapt their urban environments to the needs of an ageing population [24]. In cities such as Barcelona, action plans have been promoted that integrate citizen participation, safe mobility and inclusive spaces [25]. At the international level, cities such as Manchester and New York have implemented innovative strategies focused on accessibility, adequate housing and intergenerational cohesion [26]. These experiences reflect a common approach to active ageing based on health, inclusion, and the urban environment as key determinants of well-being [27]. The aim of these projects is to transform urban environments into spaces that respond to the needs of an ageing population through inclusive infrastructure, services, and policies that enable people to age safely, healthily, and participatively [28,29]. However, some recent studies warn that certain urban design strategies considered inclusive could have unintended effects on the functional health of older people. Gilroy and Townshend [30] point that, although the removal of physical barriers improves accessibility, it can also reduce everyday opportunities for movement, encouraging more sedentary behaviours. From this perspective, they propose rethinking how urban environments can simultaneously facilitate inclusion and encourage moderate physical activity. Elements such as walkable paths, gentle stairways, and green spaces that invite spontaneous movement are presented as key resources for promoting active and independent ageing. This reflection broadens the focus of age-friendly cities, suggesting that it is not enough to guarantee access, but that it is necessary to design spaces that promote functional mobility in the daily lives of older people.

The evidence on the role of urban nature in supporting the well-being of older adults has grown substantially in recent years. A comprehensive review by Xu, Nordin and Aini provides a valuable foundation for this discussion by identifying three key dimensions; spatial, ecological, and infrastructural, through which these spaces exert their influence [31]. Within this broader category of urban nature, urban gardens and community allotments emerge as particularly effective settings for generating health-related benefits among older populations. Physical health is one of the most consistently documented domains of improvement. Participation in gardening contributes to enhanced mobility, cardiovascular endurance, and muscular strength [32], with tasks such as planting and watering offering moderate-intensity physical activity appropriate for the capacities of older adults [33,34]. These activities have been linked to reduced risks of chronic illness, improved blood pressure regulation, and fewer general practitioner consultations [32,35]. Although gardening is not always associated with lower body mass index, it tends to encourage physical movement in a way that is enjoyable and, therefore, more sustainable than other forms of exercise [36].

Beyond physical health, gardening has been repeatedly associated with psychological well-being. The literature identifies a range of mechanisms through which gardening may alleviate stress and enhance mood. Exposure to natural settings has been shown to reduce salivary cortisol levels and support emotional regulation [35,37]. Moreover, gardening often enables participants to engage in focused, purposeful tasks that foster distraction from daily concerns, an effect explained by the Attention Restoration Theory [36,37]. Many older gardeners describe feelings of pride and satisfaction derived from watching plants grow, which reinforces a sense of purpose and emotional resilience, particularly during life transitions such as retirement or bereavement [34,36]. Cognitive improvements have also been reported, including enhanced memory, attention, and even visual function, especially when garden tasks incorporate elements of sensory and autobiographical memory [32,37,38].

Social benefits represent another key dimension. Urban gardens often serve as informal meeting places where older adults can interact with peers, share knowledge, and build supportive networks [39,40]. These social interactions are not always the primary motivation for participation, yet they consistently emerge as important outcomes, particularly in combating loneliness and fostering community cohesion [34,41]. Opportunities for intergenerational engagement, as well as strengthened ties with neighbours and partners, have also been reported, contributing to an increased sense of belonging and social utility [41,42]. In institutional settings such as assisted living facilities, community gardens have been shown to promote neighbourhood attachment and increased participation in communal events, which are particularly valuable for individuals at risk of social isolation [39].

In addition to these primary domains, other less frequently discussed but meaningful benefits include enhanced environmental awareness and prosocial behaviour. Gardening has been linked to stronger emotional connections to nature and increased intentions to engage in environmentally responsible practices [38]. Moreover, the sense of altruism and global solidarity associated with participation in community gardens, especially when linked to charitable initiatives, offers older adults an avenue for moral engagement and contribution to collective well-being [41]. In rural and ageing territories, such as South Tyrol, gardens have also been instrumental in promoting social integration and counteracting the demographic effects of depopulation [43].

Taken together, the literature highlights the capacity of urban gardens to generate layered and interdependent health benefits for older adults. The consistency of findings across geographical contexts and methodological approaches reinforces the potential of urban gardens to serve as low-cost, scalable interventions for promoting active and healthy ageing in urban settings. Equally, these studies stress the importance of inclusive design, appropriate infrastructure, and social facilitation in maximising their benefits.

In this context, this study aims to explore in depth the practice of urban agriculture in the Functional Urban Area of Valencia (FUA)as a strategy for promoting active ageing, focusing on the benefits perceived by older people through the analysis of various key factors: behavioural, personal, social, health and physical environment. Within the framework of the European Project U-GARDEN—Promoting capacity building and knowledge for the extension of urban gardens in European cities—this research qualitatively explores how participation in urban gardens contributes to the overall well-being of people aged 60 and over. To this end, a qualitative methodology based on in-depth semi-structured interviews has been chosen in order to collect personal and subjective testimonies that allow us to understand, from lived experience, the real impact of these practices on the daily lives of older people. This approach is particularly relevant in the current context of rapid demographic ageing, in which it is becoming a priority to generate empirical knowledge about the value of green spaces, such as urban gardens, in improving the well-being of older people. 

## 2. Materials and Methods

### 2.1. Study Design

The study employed a qualitative cross-sectional design, in which semi-structured in-depth interviews were conducted as an instrument capable of obtaining verbal responses to the questions posed about the research problem [44]. We adopted an epistemological approach based on realism, a theoretical position holding that language directly reflects experience [45]. We consider this approach to be the most appropriate, given that our interest was focused on the individual experiences of the participants and we started from the premise that they can express their realities effectively. In addition, the study was informed by the perspective of symbolic interactionism, which emphasises the meanings that individuals construct through social interaction and daily practices. This interpretive lens allows us to understand how older people attribute meaning to their participation in urban gardens, taking into account the subjective interpretations that emerge from their interactions and routines [46].

### 2.2. Spatial Context

The study was conducted in the territorial context of the Functional Urban Area (FUA) of Valencia, whose functional centre is the city of Valencia, located on the east coast of Spain. With a population of 830,606 inhabitants [47]. Valencia is the third most populated city in the country, after Madrid and Barcelona. According to data from the INE (2022), 28.1% of its population is over 60 years old [48]. This combination makes the FUA a particularly relevant area for researching the impact of urban green spaces on the quality of life of older people. The Valencia Functional Urban Area comprises a total of 63 municipalities, covers an area of 1741.1 km^2^ and has a population of 1,811,626, representing 68.19% of the total population of the province. In this urban and peri-urban context, interviews were conducted in various urban gardens located in municipalities of the FUA, specifically in the La Torre Urban Park (Sociópolis, municipality of Valencia), as well as in the municipalities of Meliana (l’Horta Nord), Alaquàs and Catarroja (l’Horta Sud), and in the Espai Verd in the Valencian neighbourhood of Benicalap [see Figure 1]. These gardens represent significant spaces for active ageing, self-consumption of healthy food and social interaction among older people. As they are integrated into a densely populated metropolitan region structured around a Mediterranean city, their analysis provides a better understanding of the role these green environments play in the well-being, health and daily life of the ageing population.

### 2.3. Participants and Procedure

The objective of the sample recruitment was to interview older adults who regularly participated in daily or weekly activities in an urban garden. A purposive sample of 15 individuals aged 60 or over, each of whom cultivated a plot in the urban gardens located in the aforementioned areas, was selected for the study. To recruit participants, the researchers contacted the representatives of the urban gardens by email, providing them with all the information about the purpose of the study and requesting permission to visit the urban gardens to interview the older adults. Eligible participants were individuals aged 60 years or older, residing in any of the municipalities within the FUA of Valencia, who actively used a plot in one of the urban gardens and engaged in routine gardening activities. Subjects who had difficulty verbally expressing their opinions/experiences as a result of cognitive impairment were not considered for participation in the study. Specifically, the inclusion criteria for participants were as follows: people aged 60 or over (both men and women), plot located in one of the municipalities within the FUA of Valencia; daily or weekly participation in the urban garden plot for at least 6 months; absence of cognitive impairment to answer the interview questions independently. All face-to-face interviews were conducted in the urban gardens themselves, taking advantage of the familiarity of the environment for the participants and encouraging a more spontaneous and intimate dialogue.

Participants who were willing to be interviewed at the time were given a brief verbal presentation of the study’s objective and the interview and were provided with more extensive written information. Participants who confirmed their interest in participating in the study but did not have time to be interviewed at that time voluntarily provided their contact details and received all the study information in writing for their perusal. In the latter case, the researchers contacted them to arrange an interview according to the informant’s availability. All informants signed the informed consent form for participation in the interview and subsequent use of the data. For all participants, regardless of whether the interview was conducted in person or online, the interview was audio-recorded using a recorder and transcribed verbatim. Demographic data were collected to characterise the sample and included gender, educational level, and professional profile. Table 1 shows details of the sociodemographic information of the sample.

The study sample shows a clear male predominance (13 men and 2 women), a distribution that was not intentional but rather reflects a specific sociocultural reality in the Valencian context. This difference may be related, on the one hand, to the agricultural tradition of the Valencian Community, where men have historically been more involved in farm work, which favours their continuation in horticultural practices after retirement. On the other hand, the Mediterranean welfare model, characterised by a strong family focus, has traditionally assigned older women the main role in caring for the home and family, which may limit their availability to participate in community projects such as urban gardens. This combination of factors helps to explain the lower representation of women in the sample and highlights the importance of continuing to explore the barriers to access that could be affecting the participation of older women in this type of initiative.

The age of the participants ranged from 60 to 90 years old. The 15 participants had different educational levels and professional backgrounds. The interviews continued until data saturation was reached, i.e., the topics developed from the script began to be repeated when the exercise was carried out with other people, validating that the interviews conducted were sufficient to support the research findings, insofar as the participants did not provide new information on the concept of urban gardens as spaces that promote active ageing [49,50].

### 2.4. Guide to Interview Topics

Given that the objective was to assess the impact of urban gardens on promoting active ageing, the different determinants that make up this concept were taken as a basis, including behavioural, personal, social, physical health and mental health determinants. An interview script with semi-structured questions was developed to ensure a level of order and consistency throughout the interview process. The researchers were flexible in the order in which they asked the questions to ensure the fluidity of the discussion. As for the set of questions, following Mannan and Afni [51] and Adams [52], various questions were initially used with the aim of breaking the ice and establishing a relationship and a desired level of trust with the respondent. Thus, the interviews began with an opening in which the interviewee was invited to talk to the researcher about themselves, how long they had been participating in urban gardens, why they had started participating in urban gardens, their experience, etc. Some examples of the questions the researchers asked at the beginning of the interview were: Have you worked in agriculture before or do you have family who have? What do you do now and what did you do before? How important is it for you to participate in a garden? What benefits have you gained from participating in the garden? Subsequently, the researchers introduced questions related to the determinants of active ageing. Questions were asked about physical activity, diet, psychological well-being, social support, purpose in life, retirement, peer and intergenerational learning, and the physical environment.

### 2.5. Data Collection, Measures and Ethical Issues

During the period between January and June 2024, three researchers conducted in-depth semi-structured interviews, both in person (n = 14) and by telephone (n = 1), with the aim of exploring the experiences of people over the age of 60 in relation to their participation in urban gardens and the capacity of these spaces as platforms for promoting active ageing.

The interview script used consisted mainly of exploratory questions and was designed based on a thematic outline that divided the content into analytical categories and subcategories, thus allowing for better organisation of the discourse and facilitating its subsequent analysis.

Once the interviews were conducted, the transcripts were subjected to a qualitative thematic analysis, focusing on the identification of recurring and significant patterns within the participants’ accounts. The Atlas.ti tool was used for data processing, which allowed the information to be organised, coded and interpreted in a systematic and rigorous manner. This software is particularly useful in qualitative studies due to its ability to handle large volumes of textual data, link segments of information through conceptual networks, and facilitate the of connections between emerging categories, all of which promotes a more in-depth and structured interpretation of the experiences collected [53].

Finally, the confidentiality and anonymity of the participants were guaranteed, in compliance with the ethical guidelines established by the General Data Protection Regulation (GDPR) No. 679/2016 of the European Union, ensuring that all personal information was treated securely and responsibly.

## 3. Results

The results of this study are organised into six broad analytical dimensions, corresponding to the sections explored in the interview script, which provide an understanding of how participation in urban gardens influences the promotion of active ageing in the FUA of Valencia. Firstly, behavioural factors related to physical activity and healthy eating are addressed. Secondly, personal factors linked to psychological well-being are explored. Thirdly, social factors are analysed, including education and social support. Fourthly, health and social services factors are examined in greater depth, with a particular focus on health promotion and the prevention of physical and mental illness and long-term care. Fifthly, the factor of retirement-focused life purpose is considered. Finally, factors related to the physical environment, specifically clean air, are examined… These categories highlight the multifaceted value of urban gardens as platforms capable of generating comprehensive well-being in old age by promoting active ageing (see Table 2).

### 3.1. Behavioural Factors

The first area of analysis focused on behavioural factors related to health, addressing key aspects such as physical activity and eating habits. This section explored how participation in urban gardens influences the lifestyle of older people, promoting healthier daily practices.

#### 3.1.1. Physical Activity

The testimonies collected show that most of the older people interviewed say they feel physically well or relatively well, despite the presence of certain age-related ailments. This positive perception of physical condition is often linked to their active and continuous participation in the urban garden. Even those who acknowledge feeling physically impaired find in the garden a way to counteract the sedentary lifestyle associated with retirement.

I3: *“[When asked how they feel] Fine, very well, that’s the important thing about being in a garden.”*

I9: *“I’m not 18 yet, but I’m fine. Being in urban gardens, apart from creating a symbiosis with nature, you interact with your colleagues and socialise a lot. Very good.”*

I10: *“At the moment, physically I’m not doing well, but I relax here in the countryside, because at home there’s nothing to do and now I’m retired.”*

Although some participants have relevant medical conditions, such as heart problems or high blood pressure, these do not seem to completely limit their ability to remain active, reflecting a resilient perception of physical health.

I4: *“Well, exhausted, but fine. I’m waiting for two operations, but I don’t want to have surgery […] One on my aorta and the other for a hernia …”*

I7: *“I feel perfectly fine. Apart from some heart problems and high blood pressure … but I’m fine.”*

I13: *“Lately I’ve been very worn out. My spine hurts a little and my legs hurt a lot. But I’m fine, I can cope and I’m getting myself motivated to come here [referring to the garden].”*

In general terms, urban gardens offer a form of adapted and accessible physical activity that allows older people to keep moving without the rigidity and demands of conventional sports environments. Tasks related to cultivation, such as walking between plots, digging, watering, pruning or sowing, are carried out independently and flexibly and are included in the daily routines of the interviewees, which promotes adherence and well-being, encouraging horticultural activities to contribute not only to physical maintenance but also to reinforcing the perception of autonomy and personal usefulness.

I4: *“Gardening, drinking beer, eating sandwiches …”*

I5: *“Yes, it’s more of a contemplative life than anything else. I also have a garden, but with trees. And then I also have animals and that keeps me entertained. I mean, more in the countryside than in the city, but let’s say both things go together.”*

I6: *“… I have two vices, gardening and fishing.”*

I9: *“Walking in the mornings, for example, taking my dog for a walk, I have a dog called Boira, and then when we get home I come to the urban gardens, because there’s always something to do there and I prefer to come every day.”*

I12: *“I go swimming at the weekends. The rest of the week I take my youngest grandson to school and then to the garden. […] I come here [to the garden] almost every day and now I’ve been pruning …”*

I13: *“My routine is the garden, I don’t have anything else to do, I plant a little, I keep myself busy with a few things, I go home, watch TV, and in the afternoon I come back here, walk around a bit, talk to people, and have a good time …”*

For the older people interviewed, working in the urban garden is an essential source of exercise and physical well-being, combating the sedentary lifestyle associated with retirement and maintaining an active daily routine. The interviewees highlight how this practice stimulates joint mobility, improves muscle tone and combats stiffness. At the same time, they point out the positive impact it has on mood and mental health, by offering meaningful occupation, direct contact with nature and the opportunity to spend time with other people, promoting an active and satisfying lifestyle that reinforces both autonomy and self-esteem in old age.

I1: *“Well, in the garden, I see that, for example, people who really like it and know how to do it can plant their own vegetables. Take the garden here, for example. This garden is run by [names someone] who is retired and knows a lot, so he takes good care of it and every morning he gets up and has something to do, which is to go down to his garden to see how the things he has planted are doing […] What do grandparents who have nothing to do do? They stay at home, have their coffee and then what? […] So this gives them a lot of life.”*

I2: *“Physically, it helps me not to hunch over, it loosens up my joints, it makes me feel good in my head […] and of course, I grow broccoli, leeks, strawberries, which I have now, so I eat quite well.”*

I3: *“Being here is a pleasure. There are days when you don’t feel like doing anything, but others, like today, when you say to yourself, “I need to do some exercise,” and you do it. So I think it’s perfect for people, especially at a certain age. For retirees, I think an urban garden is ideal.”*

I5: *“Well, I’m in very good physical and mental shape. Especially mentally. Because we have a good atmosphere here. […] It’s a very healthy, very relaxed atmosphere and it’s all for the best, to improve and to help people […] here we really do live together very well and everyone is happy and everyone tries to learn so that the vegetables and everything we plant and sow turns out well …”*

I6: *“The garden gives you a great sense of peace and well-being, we have a group, we have created an atmosphere and it gives you peace of mind and better health, happiness, happiness. Physically, I’m better, more relaxed, I’m not so lethargic. I don’t like television, so I come here or I go fishing.”*

I7: *“Exercising, having extraordinary colleagues … we’re actually celebrating our fifteenth anniversary. Two colleagues and I started it, and we’re very satisfied because today there are twenty of us and we have a great time. We have a goal, which is to have lunch together every day we can, and we’re doing very well.”*

I11: *“… I come here, I spend two or three hours here, even if I don’t have anything to do, because there’s always something to do if you want. […] It’s good for the plants, because plants have resources, because as soon as there’s no wind, they have roots to start growing, I mean, things like that. […] to keep you entertained for a while …”*

I13: *“Not wandering around … I get home and I sit down and my mind is racing, I think that’s the worst thing, and here I am talking to one person, then another, and now I’m going over there and there are two or three more friends of the same age […] If I were at home sitting around and my mind was racing, then my arms would be worse, my legs would be worse, my head would be worse, everything would be worse.”*

#### 3.1.2. Healthy Eating

Most of the older people interviewed said they followed a varied and balanced diet, with a predominance of fresh foods such as fruit, vegetables, legumes and fish. In many cases, this dietary pattern is directly influenced by access to produce grown in their own urban gardens, which reinforces their commitment to a healthier and more conscious diet. Urban gardening not only facilitates the self-consumption of quality food but also transforms food preferences and improves nutritional habits. Several interviewees indicated that they consume vegetables and fruit on a regular basis, in some cases as the basis of their daily diet, with a notable presence of salads and traditional dishes—such as stews and casseroles—that incorporate legumes, vegetables and animal proteins. This finding highlights the value of urban gardens as environments that promote eating habits appropriate to the needs of ageing.

I1: *“… we eat a lot of fruit and vegetables at home. Lots of fruit and vegetables. I have a daughter, aged 26, who came home one day when she was 16 and said, ‘Mum, from now on I’m not eating meat. […] And then I had to get my act together and take a quick nutrition course and look for recipes that vegetarians could eat …”*

I2: *“… we eat a fairly varied diet at home, such as vegetables, rice, noodles, soups. […] and fruit, because now it’s orange season, there are two of us at home and we eat about 18 to 20 kilos of oranges every week.”*

I6: *“I have a lot of variety [referring to what he has planted in his garden], so I mainly eat vegetables and fish.”*

I9: *“… white meat, fish and, above all, vegetables, taking advantage of the fact that we have a plot of land, so there’s always something to eat.”*

I10: *“I eat everything, I don’t have any problems with food, meat, vegetables, anything you put in front of me. It’s true that I don’t like vegetables very much, but look what we have here [points to what he has planted in the garden] […] since I’ve been here, I eat more vegetables.”*

I11: *“Well, I especially eat a lot of variety. […] I buy a variety of all fruits. I almost always have almost all of them at home. […] We usually eat stews, which are called pucheros, with chard, vegetables, chickpeas, potatoes, because I really like stews.”*


*“Tomatoes are part of my daily diet. I eat legumes at least once a week, rice, pasta …”*


Most of the older people interviewed say they regularly consume the food they grow in their urban gardens. This self-consumption not only improves the quality of their diet by incorporating fresh, varied and organic products, but also strengthens their connection with food and nature. Several participants highlight the superiority of these foods over supermarket products, both in terms of taste and nutritional quality. In addition, the consumption of these products often extends to the family or neighbourhood environment, which encourages exchange and cooperation. Some interviewees even mention that they look for new recipes to make better use of what they grow, while others list the products they consume regularly. 

I2: *“… It’s not that I eat it, it’s that my children eat it and if there’s any left over, then a relative or an acquaintance will have some.”*

I5: *“Yes, yes, yes, we do everything here and most of it is organic, so of course, the taste of, for example, a carrot from here compared to a conventional one is a world of difference, the difference in taste and smell and everything […] the products here are obviously better than those in the supermarket, without a doubt, and then there are also the nutrients we add, which are also natural nutrients …”*

I8: *“Well, that’s one of my priorities [referring to eating the food he grows]. I try to go online and always make them in different ways and not just how we’re used to, for example, aubergines, which we only eat in one way, but you can eat them in many different ways, and I try to change […] I try to use everything we grow for our own consumption, to give to colleagues, for example, I have onions and I say: take four onions from there …”*

The older people interviewed agree that their diet has improved thanks to eating food grown in the urban garden, particularly valuing its taste, freshness and naturalness compared to supermarket products. Several participants highlight that these foods give them greater confidence, as they know where they come from and how they are grown, which reinforces healthy habits and more conscious eating. Some even associate the flavours of the garden with childhood memories or greater satisfaction when eating, while others recognise a slight, though not drastic, improvement over commercial organic products. In general, food self-sufficiency and the direct link to the land emerge as key factors in this positive perception.

I1: *“I remember my father coming up with a basket full of tomatoes, cucumbers and peppers, and now when I’m given a tomato, I get that smell from my childhood. However, when I buy a kilo of tomatoes at the market, they don’t smell of anything, they’re like plastic …”*

I6: *“… well, yes, because I know what goes into the food I put on the table, I don’t add any hormones. Some people do …”*

I8: *“Yes, for the simple reason that I know what I’m eating. It’s 100% organic and the difference is, I don’t know if it’s because you grow it yourself, but it tastes better.”*

I11: *“Of course, my food is healthier, tastier, just look at the tomato plants here, look how beautiful they are, and compare those tomatoes with the ones you buy at the supermarket, they’re completely different, of course.”*

I12: *“In addition to the trend, which I already follow as part of a natural diet … when the lettuce and courgettes arrive, for example … you have to get the most out of them. […] They can be eaten puréed, grilled in slices …”*

Finally, the quotes analysed show that the interviewees perceive a clear and significant difference between the food they grow in their gardens and the food they buy in supermarkets. They particularly value the taste, freshness and safety of the products they consume themselves, highlighting the health benefits and the confidence they gain from knowing the production process. Added to this is a widespread concern for food safety and the promotion of local consumption. They also express mistrust of industrialised products and even those labelled as organic, and some point out that access to healthy food is not guaranteed for the entire population. 

I1: *“95% of what we eat comes from Morocco, where they use a lot of pesticides that are banned here […] if you look at the labels, it says “packaged in”, but they all come from either South Africa or Morocco.”*

I3: *“Oh, totally different [food from the garden and the supermarket]. You try to grow it with organic products, which is very important. Industrial products, large-scale, large-scale food production, don’t use those things because it’s more complicated […] if they’re organic, they’re very expensive and then who knows if they’re really organic. Here we know what we put into the product.”*

I6: *“Better, better. The best wealth is here, in the countryside, not in the city. And eating this food from here is 100 times better than what you get in the city. Normally, in the city, I buy things because, ecologically, I buy things, but … as little as possible, everything I can get from here I keep and have …”*

I7: *“In the city, you have to go shopping, and it’s not the same as eating something you’ve grown, raised and cared for yourself. What do you know about the things you buy? How long they’ve been there and how they’re going to be …”*

I10: *“Well, man, here we eat healthier than what they buy at the supermarket. For example, here when we have tomatoes in the summer, it’s heaven. […] Then in the summer we plant cucumbers and now, for example, I buy a cucumber from [names a supermarket] and it tastes like nothing. Now I have onions, I have beans, I have cauliflower, broccoli and lettuce, rosemary …”*

### 3.2. Personal Factors

The second area of analysis focuses on personal factors and addresses how older people perceive and experience their psychological well-being in relation to their participation in the urban garden, identifying the garden as a space that promotes mental and emotional balance.

#### Psychological Well-Being

Most of the older adults interviewed perceive themselves as having good psychological well-being, associated with tranquillity, emotional stability and satisfaction with their daily lives. Many of them highlight the positive role of the urban garden in their mental health, as it offers an active routine, contact with nature and rewarding social relationships. Although some mention experiences of stress, emotional sequelae or minor forgetfulness typical of old age, the overall balance is favourable. The garden is becoming established as a space for emotional support that promotes resilience, autonomy and psychological balance in old age.

I1C: *“Ah, I’m very well, I tell you, I took some time off because I was very, very stressed, because my job is very demanding and my body and my head were telling me that I had to stop, but now I’m great and my body is telling me to retire [I could have done so already] and devote myself to this, because the garden is filling my heart and soul …”*

I2: *“Yes, yes. And it also gives you a chance to socialise outside the garden, maybe go to the cinema or the bullring or the football. So a garden is very important.”*

I5: *“… I think being active here in the gardens also helps the mental well-being of the population, it gives you a sense of stability in terms of mental and physical health.”*

I9: *“This has many advantages […] it’s good vibes and psychologically very good. It frees you up a lot.”*

Participation in urban garden activities significantly improves the psychological well-being of older people by offering them contact with nature, active occupation and opportunities for socialisation. Many highlight that growing their own food brings them personal and emotional satisfaction, while others value it as a form of disconnection and mental relief. The space for autonomy it offers in family life and its role as a meeting place that strengthens social ties after retirement are also positively appreciated, and it is perceived as a key resource for emotional balance in old age.

E1: *“And then, being in contact with nature, for example, I go to the vegetable garden and I might spend hours walking around the garden, and I can assure you that I come back with a totally different energy, it changes everything, it relaxes me. […] It’s work that you’ve done and then you get a reward for it and then you make yourself a salad and you say to yourself, look, I made this, this came from me. […] Well, if you’ve grown it yourself, I think that on an emotional level it’s very satisfying, it gives you pleasure. […] your mind is occupied and your body is too.”*

I2: *“For me, spending time in a vegetable garden is the greatest thing because when you’re in the garden, everything else you might have, well, you don’t have it anymore, it disappears […] it’s therapy.”*

I3: *“In everything, I think physically and then emotionally, and at home it’s great because you arrive and you have your space, you’re here, your wife has hers, and then you go home and you get together, and I think it’s very good for the relationship …”*

I8: *“Well, socialising is one of the most important things I give them, and you also get some exercise, and when you get tired you talk to one person, then another … and that gives people a zest for life, especially when they’re retired.”*

I11: *“For me, it’s what I was telling you. It takes my mind off everything I have outside and when I’m here I’m focused on the four floors. Before I had this, I was without it for a while after I retired … Of course, I used to walk two or three, four hours a day, which frees you up a bit, but this is better, it clears your mind, it’s like when you have children to look after.”*

I12: *“Psychologically, it means that … what would I do at home? If every time I come here I spend a couple of hours, at least, those two long hours multiplied by 80 visits, what would I do during those 80 visits at home? What would I do? I’d probably be in the way, but here I’m doing something useful, that is, I’m looking for a production, a material production, so to speak, and this is almost a production, let’s call it spiritual or at least social …”*

### 3.3. Social Factors

Urban gardening encourages active ageing by promoting informal, intergenerational learning and providing a key space for social interaction. Older people acquire agricultural knowledge through shared experience rather than formal training, which reinforces their empowerment and collaboration. Gardening also promotes emotional well-being by fostering meaningful relationships, active routines and connection with the community.

#### 3.3.1. Education

Most of the older people interviewed lack formal training in agriculture, but have acquired knowledge through practical experience, family and intergenerational transmission, and collaborative learning with other gardeners. Some have participated in specific courses or workshops, while many highlight the value of knowledge passed on by parents or grandparents and mutual learning in the garden environment. Self-taught strategies such as reading and trial and error are also mentioned. This informal, accessible and continuous learning is perceived as an effective and adapted way of active ageing, where age becomes a strength for sharing and expanding knowledge.

I1: *“No, let’s see. I sign up for everything there is. […] For example, when they did this course on urban gardens here, I did come. […] I’ve learned a lot from the farmers here. […] Because they have, like my father, wisdom. Natural wisdom. […] They don’t teach you that at university.”*

I3: *“Well, they give courses here and what they explain is always interesting […] it’s always good. They give seminars from time to time, some courses, some talks… well, I think they’re interesting […] As for my colleagues, there are people who have much more experience because they’ve been doing it longer, and it’s always good to learn […] time makes you correct your mistakes. The first year goes by like everything else, you make mistakes, like you always have, but little by little you learn and the interesting thing is that last year I did this wrong and now I do it a little better …”*

I5: *“I’ve been evolving with all this my whole life. Especially when I was young, I adored my grandfather, and he taught me and told me […] I’ve been cultivating myself all my life, so you had to acquire knowledge through word of mouth […] so when you get together or when you get close to someone who has a lot or knows a lot or … then you always learn something, you always learn something.”*

I7: *“Well, I’ve read a book, specifically, which I’ve had for five years now, that talks about the whole subject of agriculture based mainly on the Mediterranean area. And it helped me a lot. But what has helped me the most is what I’ve learned here, because here there are people who really know.”*

I9: *“The training is … you try things out and you go along … as you talk to all your colleagues, some know more, so they explain things to you and you try them out too.”*

I11: *“Just through practice and also from the older people, it’s a chain. I remember when I was a child and I used to go with my father, and you learn from the older people about planting, the manure you use, the products … Of course, I know all that because I’ve learned it from the older people.”*

Most of the older people interviewed show a strong willingness to share their agricultural knowledge with both their peers and younger people, valuing experience as a useful source of knowledge that is worth passing on. Many are already actively involved in mentoring new gardeners, valuing teaching as a way to foster commitment, coexistence and the continuity of traditional knowledge. The importance of collective learning and the daily exchange of knowledge among peers is also highlighted, as well as the desire to bring new generations closer to these practices from an early age. This educational commitment makes the urban garden a practical, community-based space for intergenerational learning.

I5: *“For twelve years or so, even at university, I’ve always tried to teach people what I know, so I’ve tried to teach people so that they can learn about this [gardening] and do their own little things because I know it’s very relaxing and I’m delighted to be able to share my knowledge with others; that’s what I’ve lived for.”*

I6: *“People my age, if they came, yes, right away, if they committed themselves, yes, because here a lot of people think they can come and it’s like going to Mercadona, plants and the day after tomorrow you can pick them up, but here you have to be here every day …”*

I7: *“We’re already doing it, as much as we can, there’s no problem here, neither with the association members nor with outsiders. Lately, a lot of young people have joined, and we need them, and we’re here to help anyone who comes.”*

I9: *“Yes, of course, I became a grandfather two months ago, and I’m deciding that when she’s 4 or 5, I’ll bring her here, and I’d like to pass this on even more.”*

The responses collected reflect a broadly positive view of the interaction between older people and younger generations in the context of urban gardens. Most of the interviewees agree that this relationship can be hugely enriching for both sides: older people contribute experience, practical knowledge, serenity and a more settled outlook on life, while young people bring enthusiasm, vitality and renewed energy that benefits older adults emotionally. This interaction is seen as a two-way teaching–learning process that not only facilitates the transfer of agricultural knowledge, but also strengthens social ties, fosters mutual understanding and creates fairer, warmer and more motivating spaces for coexistence.

I1: *“Everyone here is retired now, and I promise you that when I talk to them, I notice that they are a wealth of wisdom. There are some who have hardly been to school, but they have a wisdom […] that they have acquired […] if young people want to learn, that’s good, because we are treating today’s youth very badly and they also have power, and there are many older people who have told me that they are willing to come with the kids [to the garden] to teach them and then let them continue on their own …”*

I3: *“Yes, the relationship is great [with the young people] and they also want to gain experience and you feel really good being able to teach something of your own to the young people who come.”*

I6: *“Well, we could give the young people experience and they could give us vitality …”*

I7: *“The relationship is built up over time and you help them with the little you know … in the long run, that’s what friendship is all about and it makes it more enjoyable for everyone to come.”*

I8: *“Well, look, intergenerational association is very important, and we have a school garden […] young people today have no idea. You ask them, “Where does this come from?” and they say, “From Mercadona” [supermarket]. That’s important.”*

I9: *“Above all, passing on experience and showing that there are people older than me who give you a sense of serenity and tranquillity, which is very commendable.”*

#### 3.3.2. Social Support

Most of the older people interviewed maintain an active social life through various weekly activities, with the urban garden being the main place for meeting and socialising. There, they not only share agricultural tasks but also experiences, learning and meaningful bonds. In addition, they complement their socialisation with activities such as hiking, fishing, swimming, visits to the market, or occasional meetings with friends. Family relationships are also important, whether through visits from children or walks with their partner.

I2: *“… I don’t go to the pub during the week, because if you go out, you come here to the garden, you walk around the garden, if you’re at home and a friend comes over or your children come over …”*

I9: *“… this is my main activity, I was also in the dance group […] Yes, my wife dances in the group and I used to sing in the group. Now, due to lack of time, I’ve given it up because I was a bit overwhelmed with too many things. […] Fortunately, I have many relationships …”*

I12: *“We usually get together once a month with colleagues from here [referring to the vegetable garden] in Alaquas. One colleague is also a teacher in Cuenca and another person [name of the person] is also from Cuenca, and we talk about things related to young people, such as food, military service, girlfriends, our lives … as if religion and politics didn’t exist.”*

I13: *“Mainly here [in the garden] and at half past five I go home and spend a little while there with her and then at half past five or six I leave and we go out for a walk. […] And at the weekends I go to the countryside, I’m in the countryside every weekend.”*

I15: *“Well, we get together with friends [in the garden], one tells a joke, another tells another, I tell another. And that’s the relationship I have, because now I was sitting with a friend who can hardly walk with [name of the person], and I feel sorry for the man. But then you get there and you spend some time with him and you talk and he tells a joke that I tell.”*

The interviews reveal that the urban garden functions as a key space for social interaction for older people, where meaningful bonds are formed and a mutual support network is promoted. Most participants confirm that working in the garden allows them to interact with other people in a natural and constant way, whether through shared tasks, everyday conversations or mutual help when faced with difficulties or doubts. These relationships not only improve their emotional well-being but also contribute to their physical and mental health by creating an environment of camaraderie, solidarity and belonging. The interaction goes beyond the instrumental; it involves trust, affection and a collective sense of community that reinforces the feeling of usefulness and active participation.

I2: *“Yes, of course, I do what I can and what I can’t do, I say, “hey, so-and-so, can you do this for me?”, and they do it […] for example, I can’t bend my back, but as there are several people watering, I say, “open it for me”, and they open it for me, and then I tell them I’ve had enough and they close it, and I’m grateful.”*

I9: *“It’s good for my health [the urban garden], I think, health, I don’t know, especially the interpersonal relationships between the “gentola”, between the good people around here, and I’m very comfortable, I’m very happy.”*

I11: *“Very good, as I said before, very good. Me and [names his friend from the next plot], it’s just what I’m telling you, there’s a relationship, I knew him from seeing him around here and I knew he was a teacher from the beginning, but now we know each other, well, several more things that I didn’t know about him and he didn’t know about me. […] There’s an incredible sense of solidarity here in the allotments, and we don’t lock up any of our small tools or anything. It’s incredible …”*

### 3.4. Health Factors and Social Services

Urban gardening is emerging as a key resource for promoting health and preventing physical and mental illness among older people. The regular physical activity involved promotes vitality, reduces sedentary lifestyles and improves mood, sleep and overall emotional well-being. Beyond its physical benefits, gardening encourages mental stimulation and social inclusion, creating a therapeutic environment that helps prevent isolation and dependence. Participants also highlight its role in long-term care, pointing to improvements in depressive symptoms and the inclusive and cooperative nature of gardens, which offer meaningful participation for all.

#### 3.4.1. Health Promotion and Prevention of Physical and Mental Illness

Participation in urban gardening is perceived by most of the older people interviewed as an important factor in improving their physical and mental health. The tasks carried out in this space provide regular physical activity, contribute to daily vitality and offer a practical alternative to a sedentary lifestyle, preventing isolation or inactivity after retirement. In addition, many point out that this activity not only strengthens the body, but also improves mood, appetite, sleep quality and overall emotional well-being. For some, the garden successfully replaces the gym or becomes a functional exercise environment through traditional techniques. The diversity of tasks, together with a healthy and multicultural environment, also enriches the experience and promotes overall health.

I3: *“Yes, yes, yes, yes, well, the physical improvement is very good, and also, when you come and spend two or three days doing physical exercise, you feel much better, you have more appetite, you sleep better, which is very pleasant.”*

I5: *“Well, there’s everything here, isn’t there? But generally, the people who are here are all well because they do what they think they should do. They don’t do any more or any less, no one is forced to do anything […] the atmosphere is cheerful, it doesn’t matter if you’re old or young, that’s how it is, it’s the atmosphere, healthy, calm, relaxed, where everyone communicates, we don’t care if you’re from Peru or China. Because here there are Chinese, Colombians, Argentinians. There are all races, colours and everything here. There are black people, white people, there’s everything here. Separate, separate… But here that’s not noticed, here that’s not categorised.”*

I6: *“Yes, I’ll say it again, instead of living a bad life in Valencia, sitting around […] well, look here. I haven’t stopped all morning, this afternoon I’m going back … […] Here, well, look, normally, from working with the machine, weeding, hoeing, sowing, you normally go with a hoe, scratching and removing weeds, you have a pretty good job within the limits of what’s available.”*

I8: *“We have a machine, a motorised hoe, but I like to do it the old-fashioned way, with a hoe. Why? Because as long as I can do it, I get exercise. If you think about it, you can do all the exercises you want here.”*

The older people interviewed unanimously highlight the positive effects that working in the urban garden has on their mental health. For many, this space represents an environment of disconnection, tranquillity and concentration, where they can take their mind off their daily worries and focus on manual tasks. The pleasure of activities such as planting, weeding or caring for plants from germination to harvest reinforces the emotional bond with the garden and keeps the mind active, as it requires constant attention, decision-making and functional memory. Likewise, the social component associated with gardening is also perceived as key to maintaining a sharp mind. Taken together, urban horticulture is presented as a comprehensive therapeutic tool that contributes to cognitive stimulation, daily enjoyment and mental balance in old age.

I2: *“Here I forget everything, I mean, I’m at work and if I have to make a ridge, I’m not thinking about the birds flying overhead, I’m focused on what I’m doing […] the countryside is wonderful, all the tasks are enjoyable, from the moment you start planting to seeing the plant grow and moving it around, checking for pests that could harm it, taking care of it because later that plant will give me a product and satisfaction when I eat it.”*

I3: *“Good, very good, it’s great for me [being in the garden] because I have a bit of tinnitus, so it’s perfect for me because there’s no noise here, which is very good […] And I’ve also shared this idea with some of my fellow gardeners and they also say that mentally and psychologically, they think it’s wonderful. I think it’s very good.”*

I6: *“I feel good about myself [in the garden]. When I come in here, it’s like when you go to work, you switch off, and when I come in here, I leave my problems behind.”*

I11: *“Mentally, the best thing, what it gives me most, is that until I leave here, my mind is focused on the four plants. If you tell someone that you can spend 4–5 h here every day […] it’s simply that you find a way to pass the time doing little things, of course.”*

I12: *“Well, here you can switch off if you have something on your mind and you arrive and say, “What do I have to do today?” Well, today I have to start removing this, or I’ll go and see if the beans have sprouted, or I’ll go and sulphate the cabbages so they don’t rot…”*

#### 3.4.2. Long-Term Care

The interviewees’ responses indicate that working in urban gardens could help reduce the risk of dependency or disability in older people, mainly due to its ability to promote physical activity, mental stimulation and emotional well-being. Some participants report improvements in depression and highlight the therapeutic value of being in contact with nature and maintaining an active routine, while others emphasise the importance of mutual support and the inclusion of people with disabilities in a cooperative environment.

I1: *“Sure, it improves [mental health problems]. I know people who suffer from depression, especially mental illness, and I know people who have been cured by going to the garden. Little by little, I think, as I said, nature works miracles …”*

I5: *“This enriches the soul, the body, the mind, it enriches everything. You come here lame and you leave running.”*

I6: *“… if a person is at home sitting down and not moving, logically they have more disabilities than someone who comes here and works. For example, next to 100 [plot number], there is a very old man, an elderly man, who walks with a stick, and you have him there working every day. That person won’t get depressed or lethargic because of that, because he’s active …”*

I7: *“For me, apart from what the fields are, I consider this a health centre and that everyone should consider coming and having the opportunity to enjoy this option that we have, which years ago we couldn’t even dream of having.”*

I9: *“Yes, for example, if it can help a person not to be dependent. To be more agile. In fact, the guys from [names an association] with functional diversity, at a certain point they may find that by coming here you set guidelines, there is an improvement in mood.”*

### 3.5. Purpose in Life

Urban gardens provide retired older adults with social connection and a renewed sense of purpose. While some faced challenges during retirement, many found gardening to be a meaningful activity that promotes active ageing and personal well-being.

#### Retirement

The experiences of retirement among the older people interviewed are diverse: while some saw it as an opportunity to reorganise their lives and enjoy new activities such as urban gardening, others found it difficult due to health problems, forced retirement or the loss of their role at work. The emotional impact of this stage depends largely on factors such as health status, the voluntary nature of retirement and the support environment. Urban gardening emerges as a useful tool for continuing physical and social activity, providing a new purpose in life.

I6: *“Well, I retired a year early because I had colon surgery, and the only thing I’ve felt is that I’ve retired and started seeing doctors […] this activity has also helped me fill the time.”*

I8: *“Well, when I retired, I found that I had to find ways to keep myself busy. It’s a bit of a sudden change, because you have routines and then you have to change them, and now I have to come here in the mornings and do things …”*

I10: *“… A person has to retire and retire, I’ve felt fine, I haven’t had any problems, I haven’t had any stress or anything like that […] this [having the garden], well, first I came to help a lady who my wife used to look after and her husband died and she had a plot there and she got me involved without meaning to and then I went to the town hall and applied.”*

The responses analysed show that participation in the urban garden after retirement was a strategy adopted by many of the interviewees to fill their time in a meaningful and active way. For most, this meant spending more time in the garden, finding in it not only a source of entertainment, but also a space that promotes well-being, usefulness and health.

I3: *“Afterwards, shortly after [retiring], I quickly realised that I liked it.”*

I5: *“Yes, this is investing in your health when you’re older.”*

I7: *“… as soon as I found out about this in the newspaper, I went to the council and they gave me the application form, I filled it in and when I had the chance I joined. For me, this has been incredibly positive.”*

I9: *“Yes, 100% [it takes up the time I used to spend at work, in the garden].”*

I11: *“I often come here, maybe I bring a banana or something, and there are several others who come here for lunch too, and you chat for a while and clear your head, you know […] that’s what I mean, spending two or three hours, even if you don’t have to do anything.”*

### 3.6. Physical Environmental Factors

Older adults perceive urban gardens as healthier and more relaxing spaces than the urban environment, highlighting their cleaner air, tranquillity and nutritious homegrown food. Despite some concerns about pollution and chemical use, most participants express a strong preference for the garden and trust the safety, taste, and freshness of the food they produce.

#### Clean Air

The analysis of the quotes shows an ambivalent perception of the air quality surrounding urban gardens. On the one hand, several interviewees clearly highlight the respiratory and well-being benefits associated with the natural environment of the garden, especially in comparison with the urban environment: they value the vegetation, the lower traffic and the cleaner air in this green space. In other cases, there is a personal need to physically distance oneself from the city to avoid stress, breathe less polluted air and improve quality of life. However, other testimonies reflect a more critical or sceptical view: mention is made of pollution from the use of chemicals in the garden itself, or a broader mistrust of the global environmental state, including water, the atmosphere and the seas. Thus, although urban gardens are recognised as spaces with relatively better air quality, some interviewees question the extent to which they can really be considered healthy in the context of widespread pollution.

I1: *“Well, sometimes yes, sometimes no, because it’s what I’m telling you. For example, sometimes I put my hand up because when they’re spraying the air, I can feel it’s polluted and you can smell it a lot. […] Well, imagine that all this is stuck to the fruit in the supermarkets and we’re eating it […] and then, when it’s more or less normal [without the smell of sulphur], well, you do notice that, of course, there are a lot of trees, a lot of green, a lot of this and that, so it’s much better than in the city.”*

I5: *“… The air here is polluted, well, here in Valencia and in all the big cities. In all the ones I’ve been to, the pollution is terrible. So I’ve bought some land outside the city, 60 kilometres away, in Albaida, where I have fruit trees, animals and everything. Otherwise, this place stresses me out, the oxygen here stresses me out a lot, it makes me nervous, it makes me neurotic. And when I’m there, I sleep really well, I feel great, I mean, you can tell in everything, I feel it in my body, in everything …”*

I9: *“100% or 1000%. It’s true that we’re still close to the town, but even so, the air is wonderful.”*

I12: *“Here where we are, yes. Because even though we’re a stone’s throw from Valencia, we’re surrounded by trees on the right and trees on the left and some amazing green areas. […] So the air I breathe is healthy.”*

I14: *“Well, no, because there’s a lot of traffic. If you go out on the street, there’s a lot of traffic, a lot of smoke. If you go to a shop [names different supermarkets], you can’t breathe properly either. You have to go out to the countryside to be able to breathe properly. I mean, you breathe better in the garden or in the countryside.”*

I15: *“That’s not true. No, everything is polluted, and it’s getting worse every day. If we look at all the old cars that pollute and the planes that pollute, it’s outrageous, and everything is polluted. Even the sea is polluted, the boats, so everything is polluted […] because the planet is in a very bad way … the thing is, we don’t realise it, especially older people, but young people are studying and know more about what’s going on.”*

The interviewees reflect a general preference for the urban garden environment over the urban environment, especially in terms of emotional well-being and tranquillity. Several interviewees clearly express that the garden represents a space of calm, relaxation and personal satisfaction, contrasting with the stress, pollution and “commotion” they associate with the city. For some, this contrast is so striking that they would prefer to live permanently in the garden. Others value the emotional component of the work done there. Despite this, there are also testimonies that minimise the difference between the garden and the city, either because the garden is very close to the city or because they do not perceive a significant change.

I2: *“I feel different [when I’m in the garden] because I come with the excitement of doing a job, I do it, and when I leave, I look at it and look at it again. It’s personal satisfaction.”*

I3: *“Well, as I said, the fresh air, let’s say, the contact with people. I also like the city, of course, I like sharing and having friends, going out with them …”*

I6: *“Well, the city is very … I don’t like it very much. Just yesterday we went to the centre and, to be honest, I felt overwhelmed […] No, I don’t like it. I prefer peace and quiet.”*

I7: *“Well, I would stay here [in the garden], yes.”*

I8: *“It’s very difficult to explain. Here today it feels like you’re in a more proper environment [in the garden]. When you’re in the city, you see all the pollution, which also reaches here, and that makes me feel uncomfortable. I’ve realised that the older I get, the less I can stand crowds; I look for peace and quiet.”*

I14: *“Well, you notice the change between being outdoors and being in the city, or in the village, wherever you live, of course you notice it.”*

## 4. Discussion

This study highlights the potential of urban agriculture as a strategy to promote active ageing, based on the benefits perceived by older people who participate in this practice. The results, aligned with previous research, show that community gardens offer a wide range of advantages for the well-being of older adults. Interview data suggest that gardening helps to preserve physical, psychological and social health by encouraging regular, moderate-intensity activity and reducing sedentary behaviours after retirement. Although these effects were not measured quantitatively, participants described gardening as a meaningful routine that keeps them physically active, contributing to the maintenance of joint mobility, muscle tone, and overall physical vitality. Gardening was not perceived as conventional exercise but rather as an accessible, enjoyable task that integrates physical movement into daily life, enhancing its long-term sustainability. These findings support the idea that community gardens contribute to an active and autonomous lifestyle, consistent with existing studies on the benefits of engaging with urban green spaces for physical and psychological health [35,54,55]. Along these lines, the results reported a positive impact of participation in urban gardens on social and emotional well-being, due to their ability to promote social contact and expand social networks, especially after the end of working life and the onset of retirement, making them a tool for promoting social participation, a sense of belonging and resilience. A recent study has addressed the importance of urban gardens as facilitators of community connections and frequent socialisation [56]. In addition, the findings indicate that community gardens have a positive impact on food consumption. Users of the plots agree on the nutritional benefits of consuming home-grown food, not only because of the continuous access to fruit and vegetables, which promote health, but also because it is fresh food. These findings are consistent with existing literature suggesting that growing one’s own food in small green spaces improves dietary practices [57,58].

The narratives collected in this study point to urban gardens as significant enablers of physical activity among older people, offering a form of exercise that is perceived as accessible, purposeful and embedded in everyday routines. Rather than engaging in structured physical activity in formal settings—often perceived as rigid or intimidating, participants describe gardening as a practice that accommodates their bodily capacities, preferences and rhythms. This aligns with findings from previous literature, which indicate that allotment gardening offers moderate-intensity activity conducive to maintaining muscular tone, joint flexibility and cardiovascular health, without the psychological or logistical barriers that can accompany traditional sports or fitness programmes [32,33]. In this regard, gardening may represent one of the most sustainable forms of physical activity for ageing populations, particularly when it is integrated into habitual routines and tied to a broader sense of purpose.

Participants consistently associated their physical engagement in the garden with improved well-being, reduced sedentary time and a more active lifestyle. These findings are supported by evidence that urban gardens contribute to improved physical function, reduced subjective health complaints and greater cardiopulmonary endurance in older adults [32,35]. Importantly, these benefits appear to be grounded not only in the physiological exertion involved, but also in the autonomy and satisfaction derived from cultivating one’s own space, following personal schedules, and enjoying the results of one’s labour. As highlighted by Hawkins et al. [36], the distinction between “doing” and “being” in the allotment environment helps explain why participants remain motivated and engaged: physical effort is not framed as exercise, but as meaningful contribution. Moreover, the results confirm the role of gardening in reinforcing a sense of agency and usefulness among older adults, helping to structure their daily routines in post-retirement life. Many participants described the garden as their main or only structured activity, offering them a space of movement, interaction and continuity. This echoes the conclusions of Mejías Moreno [34], who observed that older adults view the garden as a source of both physical maintenance and emotional reward. These functions are especially relevant in later life, when risks of isolation, passivity and decline in mobility increase. The reported experiences also resonate with evidence suggesting that garden-based activities enhance self-rated health, particularly when they are performed in a supportive and aesthetically pleasing environment [31,39].

In addition to its value for mobility and fitness, the garden is strongly linked in participants’ accounts to their dietary habits and food practices. The testimonies reflect how access to self-cultivated produce not only diversifies the diet but encourages more conscious and healthier eating choices. These results are consistent with findings from studies in both European and North American contexts, which have identified gardening as a driver of improved nutritional behaviour, particularly through the consumption of fresh vegetables, legumes, and fruit [33,40]. Several participants described how their involvement in gardening influenced their food preferences and cooking habits, often extending the benefits to their families or neighbours through informal exchange or sharing of the produce they grew. The quality and freshness of garden produce were repeatedly emphasised as superior to store-bought alternatives, both in taste and trust. This perception reflects a common pattern in the literature, where older adults report a heightened appreciation for food whose provenance is known and whose cultivation they can control [36,40]. In some cases, gardening evoked memories of rural or family-based agricultural practices, reinforcing emotional bonds with food and land. These reflections also suggest that urban gardening may contribute to a sense of food security and personal responsibility for health, while encouraging environmentally sustainable behaviours, as suggested by Jo et al. [38].

Furthermore, urban gardens are perceived by older participants as environments that promote emotional stability, autonomy, and a renewed sense of purpose in later life. These spaces appear to offer a unique form of support for psychological well-being, through a combination of structured routine, contact with nature, and social interaction. The testimonies suggest that gardening supports both emotional regulation and resilience. Participants recurrently described the act of tending to plants, engaging with the land, and sharing space with others as key elements in managing stress and cultivating a positive state of mind. These experiences closely echo the restorative functions attributed to urban gardens in studies such as those by Hawkins et al. [36] and Guo et al. [39], where allotment and community gardening are shown to provide an effective coping mechanism through sensory engagement, distraction from daily stressors, and the stimulation of positive emotions.

The sense of psychological renewal expressed in the interviews is consistent with Kaplan’s Theory of Attention Restoration [59], as highlighted by Hawkins et al. [33], suggesting that the gardens offer a cognitive “reset” that facilitates mental restoration. The comments made by participants who described arriving at the gardens feeling depleted and leaving reinvigorated reflect this dynamic, underscoring the gardens’ role as restorative landscapes where attention fatigue can be alleviated. This pattern also aligns with the findings of Zutter and Stoltz [41], who observed that older adults identified community gardens as spaces of peace and continuity, particularly valuable in the aftermath of stressful life events or broader social disruptions. Moreover, the emotional benefits reported appear to derive not only from solitary activity but from the quality of the social environment. Several participants linked their psychological well-being to the relationships formed in the garden, with some referring to the site as a space of conviviality and mutual support. These descriptions are consistent with the evidence presented by Mejías Moreno [34] and Dyg et al. [40] who point to the role of urban gardens in combating loneliness, fostering community attachment, and promoting a shared sense of value and recognition. In this sense, psychological well-being cannot be disentangled from the social capital that gardens help to generate and sustain. Crucially, the perceived emotional benefits were often described in contrast to the sedentary and isolating routines associated with retirement or widowhood. Participants noted how gardening structures their day, offers a meaningful activity, and allows them to feel useful—elements also highlighted by Wen et al. [42] and Jo et al. [38] as essential to preserving a sense of identity and creativity in later life. Urban gardens, thus, serve both as therapeutic environments and as sites of agency and self-definition for older people.

The findings also confirm that urban gardens serve as crucial social infrastructures that facilitate informal learning, social engagement, and personal meaning-making in later life. Across the interviews, participants described the garden as a space where knowledge is acquired through practice and also generously shared, creating a dynamic ecosystem of intergenerational exchange. These results align with Wen, Albert, and Von Haaren [42], who argue that gardening fosters creativity, identity, and participation, particularly when older adults build their own environments and engage in shared cultivation tasks. The data highlights that older gardeners often see themselves as custodians of traditional agricultural knowledge, acquired through family transmission, observation, and iterative trial and error. Many expressed a strong willingness to transmit this knowledge, especially to younger generations, which reinforces the findings of Mejías Moreno, who reported that urban gardens in Spain function as spaces of social cohesion and collective learning [34]. This pedagogical commitment appears deeply rooted in a sense of purpose and social utility, contributing to self-esteem and delaying the social disengagement often associated with old age [41].

In parallel, the results underscore the role of the garden as a generator of affective bonds and emotional support networks. Interactions among peers—sharing tasks, offering advice, or simply enjoying each other’s presence—reflect a form of social capital often overlooked in urban design. These insights are in line with the literature by Guo, Yanai, and Xu [39], who observed that community gardens contribute significantly to neighbourhood cohesion and psychological well-being, particularly among seniors living alone. As in Dyg et al. [40], the garden emerges as a therapeutic environment where stress is alleviated not only through exposure to nature but also via informal peer support. Concerning health promotion, the testimonies reinforce previous evidence on the physiological and emotional benefits of gardening [32,37]. Participants frequently described improvements in sleep quality, appetite, and mental clarity. Several rejected conventional gyms in favour of the garden, which they perceived as a functional, age-appropriate form of exercise that fosters autonomy. This aligns with Tharrey and Darmon [35], who identify gardening as an activity that supports cardiovascular, muscular, and cognitive health while also mitigating the risk of dependency in older age. While the health benefits are widely acknowledged, the results also reveal ambivalence regarding environmental quality. Although participants valued the air quality and tranquility of the garden environment compared to the city, some questioned the actual healthfulness of the surroundings, citing pesticide use, pollution, and global ecological decline. These concerns suggest that even highly localised green interventions must be critically assessed within broader environmental frameworks. Nevertheless, the gardens remain perceived as therapeutic refuges from urban stress, echoing the findings of Hawkins et al. [33] and Jo et al. [38] on the restorative power of nature exposure.

### Study Limitations

This study presents several limitations that should be acknowledged when interpreting its findings. As with most qualitative research, the analysis is not intended to yield statistically generalisable results, but rather to provide a situated and in-depth understanding of the meanings and practices associated with urban gardens in the context of active ageing. The sample was composed primarily of senior gardeners who had already developed a sustained relationship with their garden, which may have introduced a positive selection bias. Consequently, the views expressed may not reflect the experiences of individuals who discontinued participation, encountered barriers to entry, or developed critical perspectives on garden dynamics.

In addition, the interviews privileged the voices of older adults who actively engage in urban gardening and reported largely positive outcomes. While this provides valuable insight into the enabling potential of such spaces, the study does not incorporate the perspectives of older adults who may experience physical limitations, social discomfort, or difficulty accessing these environments. As highlighted in recent literature (Yuan and Chen), gardening can pose physical risks, such as joint strain or overexertion, if the activities are not adequately adapted to the needs of senior participants. These potentially exclusionary effects, often linked to a lack of ergonomic infrastructure or inclusive design, were not addressed in the present study [32].

Another limitation of this study is the gender imbalance in the sample, which consisted of 13 men and only 2 women. This difference in representation is not a deliberate decision on the part of the research team but rather reflects a particular social and cultural reality in the Valencian context. On the one hand, this imbalance can be explained by the deep-rooted traditional agricultural model in the Valencian Community, where men have historically played a predominant role in agricultural activities, which favours their subsequent involvement in urban gardens after retirement. On the other hand, this trend is related to the Mediterranean welfare model, characterised by a family-oriented structure in which women have traditionally assumed care responsibilities in the domestic sphere (grandchildren, spouses or dependent relatives). This division of roles reduces the time available to many older women and affects their participation in community and leisure activities, especially those that require physical presence, physical effort or prolonged dedication outside the home. Although there are no specific studies on gender inequality in urban gardens in the Valencia area, these sociocultural factors offer relevant clues for interpreting the low female presence in this research.

These limitations suggest the need for future research that incorporates a broader spectrum of participant experiences, including those who are less engaged, marginalised or ambivalent towards urban gardening. Comparative studies across different types of urban gardens—varying by governance model, accessibility, or location—could also help to identify structural factors that either facilitate or inhibit inclusive ageing and community integration. Finally, the gender imbalance reflected in this study, with a large majority of male participants, highlights the need not only to ensure greater female representation in future research, but also to conduct a more in-depth analysis of the specific barriers that limit women’s participation in this type of initiative, in order to design effective strategies that promote their inclusion. In addition, it is essential to carry out studies extended to other geographical contexts, which allow for comparison and validation of the findings obtained here, and to assess whether similar patterns of gender participation and sociocultural influence are observed in different urban and regional settings.

## 5. Conclusions

Population ageing and intensified urbanisation are global structural phenomena with significant social, health and environmental implications. An integrated approach is essential to design sustainable strategies that promote active ageing, understood as the optimisation of opportunities for health, participation and security in order to improve quality of life in old age. In this context, urban green infrastructure, and particularly urban gardens, are emerging as an effective tool for mitigating the negative effects associated with this demographic transition. 

The results obtained in this qualitative study confirm that regular (daily or weekly) participation in urban gardening activities generates significant benefits in multiple dimensions of older people’s well-being. These include increased physical activity, improved eating habits, strengthened social networks, reduced feelings of unwanted loneliness and evoked a general perception of well-being and personal satisfaction. Positive effects on physiological health, functional autonomy and emotional state are also observed, particularly in terms of reduced stress and anxiety. Consequently, urban planning geared towards the design and implementation of accessible and inclusive green spaces can play a key role in promoting environments that favour active ageing. 

The integration of these elements into urban public policies will not only contribute to the well-being of the ageing population but will also strengthen the social and environmental resilience of contemporary cities. In this regard, it is necessary to translate the potential of urban gardens into concrete policy measures. These include incorporating them into local urban planning, prioritising access for older people, particularly through accessible infrastructure, and promoting intergenerational projects. In addition, adopting a gender-sensitive approach, ensuring stable municipal funding and recognising gardens as tools for public health and social inclusion are all essential to ensuring their sustainability and long-term impact. This study not only supports the widely recognised benefits of urban gardening for active ageing but also offers new empirical insights into its multifaceted impact on the lives of older people from physical, emotional, social, and cultural perspectives. Based on the testimonies collected, new elements have emerged that deepen our understanding of this practice: participants describe how gardening helps them maintain an active lifestyle, preserve mobility and interrupt sedentary routines through regular, adapted physical activity with high biophilic value. Rather than confirming a measurable improvement in physical health, these accounts point to the role of gardening in supporting functional movement and contributing to a sense of physical vitality in later life. On the other hand, significant changes in diet are self-reported, such as horticultural self-sufficiency, consumption of fresh produce grown by the participants themselves and an emotional revaluation of these foods, which promote a healthier and more conscious diet. At an emotional level, gardens consolidate their role as spaces that give meaning and structure to life after retirement, strengthen self-esteem, and act as an antidote to unwanted loneliness. This study also highlights the value of collaborative and intergenerational learning processes, in which the exchange of traditional knowledge strengthens social ties and community cohesion. Finally, the potential of these spaces as restorative environments in the face of urban stress is highlighted, offering calm, disconnection and meaningful connections. Taken together, these findings expand the available knowledge on urban gardens, providing a comprehensive and contextualised view of their role as a tool for promoting well-being in old age.

## Figures and Tables

**Figure 1 ijerph-22-01058-f001:**
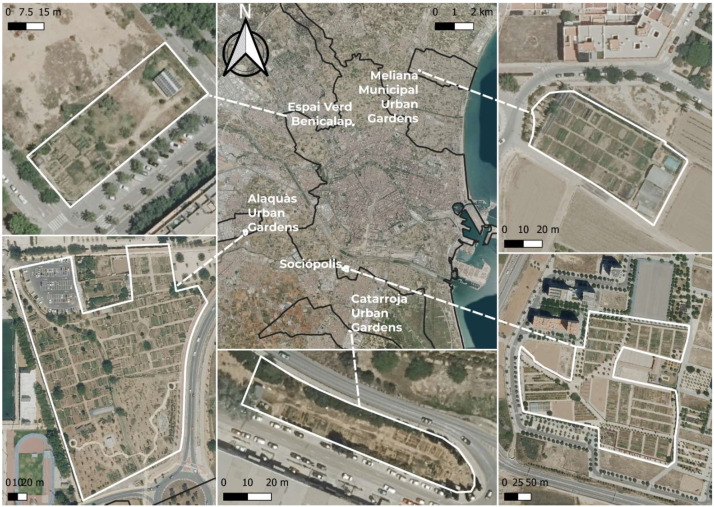
Location of the urban gardens where the interviews were conducted within the FUA of Valencia. Source: Own elaboration. Base map: Orthophoto 2024 of the Valencian Community.

**Table 1 ijerph-22-01058-t001:** Characteristics of the sample (n = 15).

N°	Garden Name	Gender	Educational Level	Professional Profile
I1	Espai verd	Female	Higher education	Commercial
I2	Meliana	Male	Primary education	Automotive trade
I3	Sociópolis	Male	Secondary education	Telecommunications
I4	Sociópolis	Male	Secondary education	Telecom and environmental services
I5	Sociópolis	Male	Primary education	Food production
I6	Sociópolis	Male	Primary education	Transport and maintenance
I7	Sociópolis	Male	Secondary education	Commercial
I8	Sociópolis	Male	Higher education	Administrative (automotive)
I9	Alacuás	Male	Primary education	Livestock and transport
I10	Alacuás	Male	Primary education	Construction
I11	Alacuás	Male	Primary education	Passenger transport
I12	Alacuás	Male	Higher education	Teacher
I13	Alacuás	Male	Primary education	Construction
I14	Catarroja	Female	Higher education	Healthcare support
I15	Catarroja	Male	Primary education	Construction

**Table 2 ijerph-22-01058-t002:** Analytical dimensions, emerging categories and key findings.

Dimension	Main Category	Key Results
Behavioural factors	Physical activity	-Positive perception of physical condition linked to active and continuous participation in the urban garden. -Reduction in sedentary lifestyles during retirement. -Adapted and accessible physical activity. -Stimulation of joint mobility, combating stiffness and improving muscle tone. -Preference for traditional gardens as a setting for physical activity (high sense of biophilia).
	Healthy eating	-Improved dietary patterns. -Transformation of food preferences. -Regular consumption of home-grown food. -Perception of higher quality of food compared to that sold in supermarkets. -Greater confidence in the food consumed thanks to knowledge of its origin and cultivation process (evaluation of taste, freshness, smell and safety of products). -Evocation of childhood memories after consuming products grown in the garden. -Greater satisfaction when eating. -Concern for local consumption. -Food vegetable self-sufficiency. -Pleasant gastronomic experience. -Source of healthy and satisfying nutrition. -Distrust of industrialised or organic products sold in supermarkets.
Personal factors	Psychological well-being	-Routine activity. -Emotional support space (promoting resilience, autonomy and psychological balance) -Personal and emotional satisfaction from growing food. -Strengthening social ties with people in the garden, replacing the social void left by retirement.
Social factors	Education	-Acquisition of knowledge through cultivation practices. -Transmission of family knowledge (respect for traditional knowledge) -Intergenerational learning (two-way teaching-learning process) -Collaborative learning among users. -Opportunity to share agricultural knowledge and gain personal recognition. -Support for new gardeners -Daily exchange of knowledge among peers.
	Social support	-Main meeting and social space -Building meaningful relationships between users. -Reducing unwanted loneliness. -Building mutual support networks. -Developing values: empathy and solidarity.
Health factors and social services	Health promotion and prevention of physical and mental illness	-Increased daily vitality. -Improved mood, appetite and sleep quality. -Gardening as a valuable alternative to limited space or a replacement for more rigid or unhealthy leisure and free time activities. -Multicultural environment (promoting cultural exchange, reducing prejudice and stereotypes, and fostering social cohesion and the inclusion of at-risk groups). -An environment for disconnecting, finding peace and concentration (performing consecutive tasks: planting, sowing, watering, cultivating, weeding, pruning, harvesting).
	Long-term care	-Reduced risk of dependency or disability (promotion of physical activity). -Therapeutic value with improved mood. -Promotion of mental stimulation.
Purpose in life	Retirement	-New purpose in life after retirement. -Urban gardening as a tool for continuing physical and social activity. -Strategy for spending time in a meaningful and active way.
Physical environmental factors	Clean air	-Ambivalent perception of air quality. -Context of widespread pollution (urban garden) -Space for calm, relaxation and personal satisfaction, contrasting with the stress, pollution and ‘commotion’ associated with the city.

## Data Availability

The data that support the findings of this study are available from the corresponding author upon reasonable request.

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
