# Peer review of "Cultivating Well-Being: An Exploratory Analysis of the Integral Benefits of Urban Gardens in the Promotion of Active Ageing"

_ijerph, 2025, doi:10.3390/ijerph22071058_

Round 1
Reviewer 1 Report
Comments and Suggestions for Authors
A good topic, see attachment for details

Author Response
COVER LETTER
Reviewer 1
Manuscript review for the International Journal of Environmental Research and Public Health, entitled "Cultivating well-being: An exploratory analysis of the comprehensive benefits of urban gardens in promoting active ageing.
"This paper used interview notes with 15 older adults aged 60+ in Venice Spain to understand the benefits of urban gardens for retired older adults. Their findings suggest that urban gardens provide 6 aspects of benefits: physical, emotional, psychological, social, nutritional and mental. The quotes are interesting, but the paper is not ready for publication. Here's why:"
Comment 1: "The conclusions are not new. Before readers read the article, they can already expect the results. The findings presented are not contextualised to allow for contextualised meanings of gardening. Each individual interested in gardening may have an individual story, perhaps due to boredom after retirement, perhaps due to widowhood, or perhaps simply due to intergenerational transmission of knowledge and interest in gardening. Very little is known about the social background of each interviewee. Moreover, there were 13 male and only two female participants. Why is this a typical scenario in Spain? In the neighbourhood where I live in the US, the proportion of gardening participants is 70% higher for women than for men. I am curious to know why there were so few women interviewed (informal care)".
Response: we would like to thank you firstly for this first comment regarding the contextualisation of the results, we have rephrased the information to make it more clarifying (section 2.2 Spatial Context, page 6 - lines 275-296) by including a figure that spatially locates and characterises the gardens analysed (see Figure 1) and a few lines about where the interviews were conducted (section 2.3 participants and procedure, page 7 - lines 318-320), in order to reinforce that contextual dimension that you are missing. Secondly, while it is true that some general benefits of urban gardening can be anticipated, the meanings that emerge in this study are deeply linked to the concrete and situated experience of older people, which gives the results a distinctive character. In this sense, urban gardening is not only presented as a healthy or recreational practice, but as a multifunctional strategy for the promotion of active ageing, in which symbolic, emotional, social and nutritional dimensions converge. This study allows us to interpret horticultural activity as a form of re-signification of post-retirement time, where cultivation provides a purpose in life and the re-emergence of daily routines. At the same time, it is configured as an act of daily resistance in the face of food distrust, in a context marked by the perception of deterioration in the quality and traceability of industrialised food. In addition, the garden is revealed as a space for reconstructing identity, where the elderly recover agrarian knowledge, revive biographical memories (such as the evocation of childhood flavours) and reaffirm their active role within the community. Another novel contribution of the study is the way in which urban gardens generate specific relational dynamics promoting the generation of social capital: from spontaneous cooperation between neighbours to the intergenerational transmission of knowledge, as well as the creation of social networks capable of mitigating the problem of unwanted loneliness associated mainly with this population group. An explicit concern for unequal access to healthy food is also documented, which connects everyday garden practices to broader discourses on food justice and urban sustainability. These elements, far from being generalisations, are empirically grounded findings and contextualised in a specific Mediterranean urban setting. This perspective not only enriches the understanding of the phenomenon, but also broadens the usual analytical framework from which urban gardens are approached, giving this study an original and necessary contribution to the field. Thanks to your comment, we have considered it pertinent to clarify and expand on these issues in section 5 (Conclusion), where the original contributions of the study are highlighted with greater emphasis (page 29 - lines 1356-1385). Finally, in relation to the gender imbalance in the sample (13 men and only 2 women) we would like to explain, despite the non-existent documentation focusing on gender inequality in participation in urban gardens in the context of Valencia, what this is due to in our experience. First of all, the distribution was not intentional, but reflects a specific social and cultural reality of the context in which the study was carried out. Thus, one of the differences in participation between men and women is linked to the traditional agrarian context of the Valencian Community, where men have historically played a predominant role in agricultural activities. Some of the participants interviewed had worked in agriculture throughout their lives or had a strong family link to this productive environment, making them more willing to continue these practices in retirement through urban gardens. On the other hand, we consider that this difference is also part of the Mediterranean welfare model, characterised by a markedly familistic nature, in which women have traditionally assumed the responsibilities of caring for the family. This gender role distribution has historically limited older women's access to leisure activities or community participation, such as urban gardening, especially when they involve travel, physical tasks or prolonged time away from home. Thus, many older women, especially in urban and peri-urban Spanish settings, are still actively involved in informal care work, such as caring for grandchildren or dependent family members, which conditions their available time and reduces their participation in community projects that require commitment and continuity. Due to the relevance of this last comment on the unequal representation between men and women, we have seen it necessary to include a paragraph mentioning this in section 2.3. Participants and procedures (page 8 - lines 335-345) and in the limitations section (page 28 - lines 1308-1322 and lines 1329-1336).
Comment 2: "The quotations are too many; some of the quotations are repetitions of the same or similar themes or meanings. I suggest a significant reduction in the number of quotes included in the article".
Response: We sincerely appreciate your observation regarding the number of citations in the manuscript and fully agree with you that the abundance of citations may have led to unnecessary repetition or made the text difficult to read smoothly. We have therefore carried out a careful review of the set of quotations and made a significant reduction of the selected material. In this process, we prioritised those quotations that offered greater expressive richness, argumentative clarity or distinctive content, and eliminated those that were redundant, underdeveloped or reiterative in meaning. In total, 51 quotations have been deleted. We thank you again for your suggestion, which we consider very valuable in improving the final quality of the article.
Comment 3: "Some of the sub-themes are redundant. For example, 3.2.1 and 3.6.3. have very similar content. I would suggest that you merge them with the same topic, but describe the differences, if you want to highlight any."
Response: We very much appreciate your comment and understand the confusion. In this sense, sub-section 3.2.1 refers to psychological well-being, while 3.6.3 deals specifically with healthy eating. Therefore, both sub-themes respond to different dimensions within the analysis of the impact of urban gardens on the lives of older people and on the promotion of active ageing. However, we understand that your comment was in fact referring to sections 3.1.2 (Healthy eating) and 3.6.3 (Healthy food), which did share similarities in both content and approach. In this respect, we fully agree with you, as there was indeed duplication between the two sub-sections. In response to your suggestion, we have unified both contents in a single subsection, maintaining a coherent structure and highlighting the most relevant differences in each analysis carried out in the subsection. This modification can be found in the current subsection 3.1.2, which contains the consolidated analysis, specifically on page 15, lines 611-639. We have also proceeded to modify the results table by deleting the row related to section 3.6.3 and reformulating the row related to sub-dimension 3.1.2 (page 10, table 2).
Comment 4: "In your discussion section, you state that gardening improves physical well-being (p.23). There is no citation linking this claim to the fact that a participant has improved their physical well-being by gardening. Perhaps I can claim that gardening "promotes" physical well-being. I fully agree that gardening promotes physical, emotional and social well-being. But to claim that physical well-being "improves", you may have to provide a more quantitative measure of this improvement."
Response: We would like to thank you for the relevance of this comment as you are absolutely right to point out that claiming that gardening "improves" physical well-being would require more quantitative evidence or an explicit statement from the participants. After carefully reviewing the text and transcripts, we have found that while there are no verbatim quotes documenting an objective physical improvement, there are multiple accounts linking gardening to a more active routine, body movement and a reduction in sedentary lifestyles, making it possible to more accurately state that the activity "promotes" physical well-being in older people. For this reason, we have modified the wording of the corresponding sections in the discussion, replacing the term "improves" with a more appropriate formulation such as "promotes physical well-being", in line with the qualitative contributions obtained (page 29 lines 135-1385).
"Overall, I find the topic interesting, but the article needs revision and polishing before it can be considered for publication."
Finally, we would like to express our sincere thanks for the care and thoroughness with which you have reviewed our manuscript. We trust that the modifications made in response to your valuable comments have contributed to a significant improvement in the quality of the study. We also cordially invite you to have a final reading of the text, as, in addition to the changes suggested by you, we have made some further adjustments in order to refine and clarify the content even further.
Reviewer 2 Report
Comments and Suggestions for Authors
A useful reference on the integration of nature with urban design for healthy ageing is: Gilroy, R., & Townshend, T. (2025). Avoiding ‘bungalow legs’: active ageing and the built environment. Journal of Urban Design, 30(2), 143–152.
Avoid using the term "elderly"; use “older adults” instead.
The paragraph in the introduction on ageism is unclear. While the topic is interesting, it does not seem well connected to the surrounding content. Consider linking it more directly to your main argument—for example, by stating that ageism is reflected in the design of cities and contributes to environments that are not age-friendly. Otherwise, it risks appearing out of place.
The main focus of the paper only becomes clear on page 4. It would be helpful to introduce it earlier with a short sentence, perhaps a couple of pages before.
The introduction is informative and clearly written.
In addition to realism, you might consider mentioning symbolic interactionism, as it could provide additional theoretical depth and align well with your analytic aims.
Please specify the locations where interviews took place—e.g., in the allotments, participants' homes, or at the university.
The sample is predominantly male. A paragraph explaining this would be useful, along with a discussion of whether this reflects the broader context of Valencia, Spain, or wider trends (e.g., the tendency of older men to be more involved in allotments, as seen in the UK). This point is important for interpreting your findings and informing policy suggestions.
The table is particularly effective and adds clarity.
Although the manuscript includes a rich set of quotes, at times it reads as a list. Consider removing quotes that do not add new insight, or grouping them under a more interpretative paragraph to enhance coherence.
Typo: Health factors and social services (añadir breve introducción???)
—please revise or clarify this.
In the limitations section, also note that the sample is small, predominantly male, and limited to Valencia.
In the conclusions, the policy implications are somewhat abstract. Consider suggesting more concrete measures—for example, expanding access to allotments for older adults, ensuring they are free or subsidised, and improving accessibility.
Author Response
COVER LETTER
Reviewer 2
Comment 1: "A useful reference on integrating nature into urban design for healthy ageing is: Gilroy, R., & Townshend, T. (2025). Avoiding 'bungalow legs': active ageing and the built environment. Journal of Urban Design, 30(2), 143-152."
Response: thank you very much for making us aware of this study. We sincerely appreciate this valuable bibliographic suggestion as we agree on the relevance of the Gilroy and Townshend (2025) article to enrich the approach to active ageing and the relationship with the built environment. In response to their comment, we have incorporated this reference in the introduction of the manuscript, right after addressing the WHO Friendly Cities initiative. In doing so, we extend the reflection on how certain urban designs, although inclusive in terms of accessibility, can have undesirable effects on the functional health of older people if daily opportunities for movement are not integrated, can be found on page 3, lines 127-137.
Comment 2: "avoid using the term "elderly"; use "older adults" instead".
Response: We sincerely appreciate your comment about the use of the term "elderly". You are absolutely right, we recognise that it is a term that can have negative or outdated connotations and that is the last thing we intend to do. We have therefore replaced it throughout the manuscript with the term "older adults", which we consider to be more respectful, accurate and in line with current recommendations in the field of gerontology and social research. The corrected terms can be found on page 2 - line 53, and on page 27 - line 1249. Thank you for your valuable suggestion.
Comment 3: "The introductory paragraph on age discrimination is unclear. Although the topic is interesting, it does not seem to be well related to the surrounding content. Consider linking it more directly to your main argument, for example by stating that age discrimination is reflected in the design of cities and contributes to environments that are not suitable for older people. Otherwise, you risk appearing out of place."
Response: Thank you for your comment on the paragraph on age discrimination. We have reworded the text to improve its coherence with the central argument of the article, explicitly linking ageism to urban design and its effects on the lives of older people. The new version makes a clearer transition between the stereotypes associated with old age and the need for inclusive urban environments, thus reinforcing the connection between social stigma, active ageing and urban planning.
Comment 4: "The main focus of the article is only clear on page 4. It would be useful to introduce it earlier with a short sentence, perhaps a couple of pages earlier".
Response: we would like to thank you for your comment which undoubtedly improves the clarity of the manuscript, as we do recognise that the main focus of the article could be introduced earlier to better guide the reading. We have therefore incorporated an introductory sentence at the end of the first paragraph of the Introduction (page 2, lines 55 and 57), where the main objective of the study is explicitly framed. This sentence summarises that the article focuses on exploring how urban gardens impact on the holistic well-being of older adults from a qualitative perspective, thus facilitating a better understanding of the main thread from the beginning of the text.
Comment 5: "The introduction is informative and clearly written".
Response: We sincerely thank the reviewer for his positive comment on the introduction to the article. We especially appreciate his acknowledgement of the effort made to present the context in a clear and understandable way. This kind of feedback encourages us to continue working with rigour and enthusiasm in the development of future research that contributes to scientific knowledge from an accessible and well-structured writing.
Comment 6: "In addition to realism, you might consider mentioning symbolic interactionism, as it might provide more theoretical depth and fit well with your analytical goals.
Response: Thank you for your valuable suggestion regarding the inclusion of symbolic interactionism as a complementary theoretical framework. We have incorporated this perspective in section 2.1 "Study design", recognising that symbolic interactionism offers a key interpretative depth for understanding how older people construct meaning from their participation in urban gardens. This inclusion allows for a richer look at the symbolic and relational processes that mediate their everyday experience, thus strengthening the coherence between the methodological approach and the analytical objectives of the study (Page 6, lines 257-262). The reference to Sucre González and Cedeño González (2019) has been added to substantiate this theoretical input
Comment 7: "Specify the locations where the interviews were conducted, e.g. in the orchards, in the participants' homes or at the university".
Response: We appreciate your comment and agree that it is important to adequately contextualise both the setting of the study and the conditions in which the interviews were conducted. In response, we have expanded section 2.2 "Spatial context" (page 6, lines 275- 292), which describes more precisely the territorial framework in which the research was carried out. The study is situated in the Urban Functional Area (AUF) of Valencia, a diverse and densely populated metropolitan region that groups 63 municipalities and is home to more than 68% of the province's population. In addition, we have incorporated a figure (Figure 1, page 7, lines 293-296) that spatially locates and characterises the urban allotments analysed, in order to reinforce this contextual dimension. We have also added an explicit mention indicating that the interviews were carried out in the gardens where the older people participated, thus allowing the natural environment of interaction to be observed and reinforcing the qualitative richness of the study, section 2.3 participants and procedure (page 7, lines 318-320).
Comment 8: "The sample is predominantly male. It would be useful to include a paragraph explaining this, along with an analysis of whether this reflects the wider context of Valencia, Spain, or more general trends (e.g. the tendency for older men to be more involved in urban gardens, as seen in the UK). This point is important for interpreting their findings and informing policy suggestions."
Response: we thank you for your comment on the gender imbalance in the sample, an issue on which you fully agreed with one of the comments made by reviewer 1, and which we consider to be very relevant for interpreting the results and better understanding the social context in which the study was carried out. In this way, we would like to let you know that the predominant participation of men (13) as opposed to women (2) was not a deliberate methodological decision, but rather responds to a specific social and cultural reality in the Valencian environment. From our fieldwork experience, we identified that this pattern may be related to the traditional agrarian context of the Valencian Community, where men have historically played a more active role in agricultural tasks. This life trajectory influences their greater willingness to maintain horticultural practices after retirement, taking advantage of urban gardens as a continuity of their previous experiences. On the other hand, this difference can also be explained by the Mediterranean welfare model, characterised by a strong familistic component, in which women have traditionally assumed the tasks of caring for the home and family. This gender role structure has limited their participation in community spaces, especially when they require sustained commitment or involve time away from home. In fact, many older women today continue to be actively involved in informal care, such as accompanying and caring for dependent family members or caring for grandchildren, which reduces their possibilities for participation in initiatives such as urban gardens. Although there are no specific studies focusing on gender inequality in urban gardens in the Valencian context, these considerations allow us to interpret the gender bias of the sample from a contextual perspective and not as a methodological flaw. Given the importance of this issue, we have added a specific paragraph in section 2.3. Participants and procedure, in order to get a first glimpse of the reasons for this lack of female representation (page 8, lines 335-345).
Comment 9: "The table is particularly effective and provides clarity".
Response: Thank you very much for your comment. We are glad to hear that the table has been particularly effective for you. We have tried to synthesise the main findings of the study in a clear and orderly way and your feedback confirms that this tool has contributed positively to the understanding of the content. We are very grateful for your appreciation and hope that future readers will also appreciate it.
Comment 10: "Although the manuscript includes a large set of quotations, it sometimes reads like a list. Consider deleting quotations that do not add new data or grouping them into a more interpretative paragraph to improve coherence."
Response: We sincerely appreciate your observation regarding the extensive use of quotations throughout the manuscript. We fully agree with you that the text could become too enumerative in tone with the insertion of so many quotations, affecting the flow of reading and the interpretative coherence of the academic discourse. In response to your suggestion, we have carried out a detailed review of all the embedded quotations. As a result, we have removed a total of 51 quotations that we considered to be repetitive, irrelevant or lacking in added value, and regrouped subsections 3.1.2 (Healthy eating) and 3.6.3 (Healthy food), which shared similarities in both content and approach.
Comment 11: "Typographical error: "Health factors and social services" (add a short introduction?). Please revise or clarify this point".
Response: We would like to thank you for pointing out this error as it was an internal note that we forgot to remove in the submitted version of the manuscript. We have removed the note "add a brief introduction" from the section under 3.4. Health factors and social services", (page 20, line 878).
Comment 12: "In the limitations section, also note that the sample is small, predominantly male and limited to Valencia".
Response: again, we are grateful for this observation and of course, you are absolutely right to point out that the limitations section should include an explicit mention that the sample is small, predominantly male and located exclusively in the Valencia area. We have proceeded to expand and qualify the limitations section by incorporating this information and stressing the need to carry out future research with larger, gender-balanced samples and extended to other geographical contexts in order to contrast the findings obtained (page 28, lines 1308-1322 and lines 1329-1336).
Comment 13: "In the conclusions, the policy implications are somewhat abstract. Consider suggesting more concrete measures, e.g. expanding access to urban gardens for older people, ensuring that they are free or subsidised, and improving accessibility".
Response: Finally, we would like to thank you for this last comment, however, we would like to point out that the main aim of this article has not been to develop detailed policy recommendations, but to focus on understanding the experience of older people participating in urban gardens from a qualitative and exploratory perspective. In relation to your suggestion, we have incorporated a brief mention of possible guiding measures at the end of the conclusions section (page 29, lines 1356-1385), in order to provide an initial connection between the study's findings and their potential applicability in the field of urban design and local policy. It should be noted that we are currently working on a complementary research project focusing on the governance of the urban gardens in the Valencia Functional Urban Area (FUA). This work in progress will focus on the analysis of management models, actors involved and regulatory frameworks, and will allow us to deepen the design of more specific and grounded policy recommendations.
Finally, we would like to thank you for the time and dedication you have invested in reviewing our work. Your input has been particularly helpful in identifying key areas for improvement, and we are confident that the revisions incorporated have greatly enriched the clarity and robustness of the manuscript. We would also like to take this opportunity to invite you, if you consider it appropriate, to make a final revision, as in addition to your valuable recommendations, we have made some additional improvements that strengthen the overall coherence of the text.
Round 2
Reviewer 2 Report
Comments and Suggestions for Authors
The authors have done a wonderful job!